# Eating, Drinking, and Swallowing Difficulties: The Impacts on, and of, Religious Beliefs

**DOI:** 10.3390/geriatrics7020041

**Published:** 2022-03-30

**Authors:** Paula Leslie, Judith Broll

**Affiliations:** 1Center for Bioethics & Health Law, University of Pittsburgh, Pittsburgh, PA 15260, USA; 2NHS UK COVID Vaccination Programme, Lancashire Teaching Hospitals NHS Foundation Trust, Preston PR2 9HT, UK; 3Royal College of Speech & Language Therapists, London SE1 1NX, UK; judith.broll@rcslt.org

**Keywords:** religion, belief, eating, drinking and swallowing, dietary law, culture, dysphagia

## Abstract

Eating, drinking, and swallowing (EDS) are fundamental to the biomechanical model of the body. They are the processes by which the body obtains fuel essential for existence but are so much more than this mere function. What, when, and how we eat, with whom, even what we do not eat, and when we do not eat, are not physiological restrictions. The Equality Act 2010 prohibits discrimination of patients based on a list of protected characteristics, including religion. There is a paucity of literature addressing religion and EDS issues despite most religions having laws regarding food sourcing, preparation, consumption, and fasting. The diverse perspectives of our patients may influence engagement with services unless we appreciate the significance of the interplay of EDS and religious belief. Our paper addresses religion and EDS with a focus on the activities that lead up to food or drink consumption. Religion, as with many important aspects of humanity, is a highly individual experience. Thus, we need to establish what is important to each person that we deal with, whilst using general knowledge of a religion to guide us. An informed multidisciplinary team including stakeholders from chaplaincy services is critical for optimal patient care.

## 1. Introduction

Eating, drinking, and swallowing (EDS) are fundamental to the biomechanical model of the body in health care. They are the processes by which the body obtains fuel essential for existence. As with breathing, we barely notice their working until something goes wrong. The fields of nutrition, gastroenterology, and swallowing disorders (narrowly termed dysphagia) have become major areas of scientific and clinical focus. Eating and drinking may also be disrupted in mental health conditions such as anorexia nervosa, binge eating disorder, and bulimia [1], but such conditions are beyond the remit of this paper. We have started to attend to the psychological impact of disrupted EDS and the effects on our social interactions. Eating and drinking are so much more than merely the functions of bringing food into the body for ingestion. What, when, and how we eat, with whom, even what we do not eat, and when we do not eat are matters of choice and cultural identity rather than physiology. Even with the restrictions of food intolerances, humans still have choice for this life-essential activity, unlike with breathing. Anthropologists have been studying these issues for decades because they come from the study of people rather than biomechanics [2,3].

For years the typical hospital intake form has had a section on religion—Why? We might ask. Historically, it may have been to ensure that the right religious leader could be contacted if the end of life was approaching. The Equality Act 2010 prohibits direct or indirect discrimination of people based on a list of protected characteristics, one of which is religion, hence the collection of data on all protected characteristics. More recent, but still over a decade since it was published, is “*Religion or belief: a practical guide for the NHS*” [4]. This guidance addressed the importance of understanding individuals’ beliefs to provide appropriate clinical care, and respectful employment, within the NHS [4]. Diverse perspectives of our patients regarding health and illness explains why they may not engage with support services or accept our well-meaning recommendations [5,6].

Although the need to collect data around religion as a protected characteristic is almost mandated within healthcare settings, there is strikingly little reference to this in relation to EDS in literature, guidance, or healthcare more widely. Religion tends to be subsumed within the sub-heading of the patient’s culture. In the recently published *Report of the Independent Review of NHS Hospital Food* (2020), there is no direct mention of any potential dietary requirements related to religious observance [7]. Discussions with the Chair of the NHS Food Review team have indicated that subsequent implementation working groups are addressing this issue [8].

Our paper will address religion and EDS with a focus on the activities that lead up to food or drink before it enters into the esophagus, where voluntary control is much less. We are concentrating on religion rather than spirituality in this short paper, in line with the requirements of the Equality Act for protected characteristics. This does not undermine the essential requirement in health care to address the spiritual needs of all our patients as we do the physiological. Bear in mind that religion, as with many important aspects of humanity, is a highly individual experience. Thus, we need to establish what is important to each person that we deal with, whilst using general knowledge of a religion to guide us.

## 2. Eating and Drinking Is More Than “Dysphagia”

Dysphagia is defined as a disorder of the swallow process, covering entry of the food or drink into the mouth through to its arrival into the stomach [9]. Broadly swallowing has been described as oral, pharyngeal, and esophageal, although these physiological boundaries do not represent the complexity of the process in real life. Voluntary control is clearly dominant as we bring food to our mouths, chew it, and move it to the back of the tongue. The change to an involuntary act occurs as the material passes into the pharynx and a conditioned motor response takes over. As material passes below the level of the upper esophageal sphincter, the voluntary aspect is almost completely over.

Dysphagia is a symptom of a clinical condition and thus treatment should address underlying causes first. Interventions to support people with EDS difficulties include texture modification, positioning, surgery, environmental factors, assistance to feed, restricting food/drink types, and alternative routes such as feeding tubes [10]. EDS difficulties are not due to ageing but are more prevalent as we age due to increased co-morbid factors, side effects from increased medication need, and decreased functional reserve [11].

Eating and drinking are so much more than physiological parts of the process, and they are directly impacted by an individual’s cultural framework. For people with religious preferences, there are many aspects of what is appropriate to eat or drink, how things are prepared, served, when not to eat, and who to eat with. Despite much literature acknowledging the importance of religion in health care, there is a paucity on the life-critical issue of EDS. A search of the terms religion and dysphagia (MeSH term) in two major clinical databases revealed twenty articles in PubMed and five in CINAHL, of which most were only tangentially related to EDS. There is more literature about porcine elements of medicines than there is on how to align religion and impairments of the activities of EDS, which affect significantly greater numbers of people.

## 3. Religion and Faith

Sir William Osler was a controversial medical educator in the late 1800s. He pushed for students to venture out of the lecture theatre, spend time with patients, and to listen to the patient in order to identify diagnoses. Perhaps more than just medical diagnosis was his understanding of the bigger picture with a patient. He was invited to write for the first volume of the British Medical Journal on the topic of *faith*:

“*NOTHING in life is more wonderful than faith—the one great moving force which we can neither weigh in the balance nor test in the crucible*.” [12] (p. 1470).

Osler described three manifestations: faith in the unseen, which aligns with religion, but also faith in those with whom we interact, and faith in oneself. Each of these is important in heath and illness. For the purposes of this paper, we will use the term religion to include any faith-based construct. In terms of the increasing move to support a diverse population, it is vital to recognize the need to identify the more profound facets of culture:

“*Whereas folk culture encapsulates dancing, games, and cooking, deep aspects of culture are not visible and can include patterns of thought, child rearing practices, religion, motivation to work, and cause of disease*.” [13] (p. 485).

Healthcare professionals are called upon to develop their cultural appreciation and humility in understanding their patients’ and families’ perspectives [14,15,16]. We prefer the terms cultural appreciation and humility rather than *competence* to reflect the values of ongoing interaction. Included within this are reflections on religious perspectives. A recent guide to migrant health for all healthcare professionals provides succinct guidance which we feel is applicable to all [17]. The debate about providers sharing their beliefs is complex and beyond the remit of our paper, but the GMC captures an appropriate stance:

“*You must not express your personal beliefs (including political, religious and moral beliefs) to patients in ways that exploit their vulnerability or are likely to cause them distress*” [18].

Almost a century after Osler and faith, the NHS addressed religion in a comprehensive report offering practical guidance in terms of care of the patient and support of the workforce [4]. The issue of how to address impairments in EDS is not addressed directly but there is a clear statement regarding the patient served by geriatric services (our emphasis):

“*Patients should always be asked to state their dietary needs; nutrition is an essential element in the treatment and recovery of patients, and patients could refuse food if it does not meet the requirements of their religion or belief. **This is especially relevant in older patients**, who may not indicate their needs unless they are asked, or in those who fear they are likely to die and are therefore even more observant in their religious practice at the time. There is a risk that the refusal of food may be attributed to a loss of appetite, leading to poor nutrition if the real reason for refusing food is not established.*” [4] (p. 25).

Care of the patient to include broad factors is increasingly seen in our move to holistic care. The range of factors to consider is growing, and the role of the chaplain is acknowledged [19]. Historically, the concept of a chaplain grew from Christianity and the military, but as far back as the First World War, there have been chaplain Rabbis and, more recently, chaplain Imams. The current chaplaincy provision in the UK extends the care that chaplains provide to come from the broadest range of religion or belief (including specific reference to lack of religion or belief) [19] (p. 5).

Palliative care and care towards the end of life are areas where we traditionally see the important role of the chaplaincy service. The recent guidance for *Care of the adult patient at the end of life* from the European Society for Medical Oncology identifies the chaplain as a core member of the interdisciplinary team due to the importance of the spiritual component of health care [20]. We propose that understanding the spiritual needs of all patients and families is integral to good health care.

## 4. Aligning Religion and Impairments of EDS

All religions have tenets regarding the preparation and consumption of food and drink, from particular substances, to preparation, to avoidance, to community participation. The Hindu Goddess of food and nourishment Annapurna was a manifestation of the Goddess Parvati who was responsible for the material world. Her husband, Lord Shiva God of the spiritual, felt that his world was more important. Annapurna demonstrated that the spiritual and material worlds are both needed to maintain balance for humanity [21]. In Roman Catholicism, wine and wafers become the blood and body of Jesus Christ: they are transubstantiated during the process of consecration in the sacrament of communion [22]. Islamic jurisprudence addresses dietary issues for halal (allowed) and haram (banned) foods and appropriate methods of preparation. Judaism has a clear set of dietary laws (kashrut), including food preparation/mixing [23]. Many of the world’s religions have common guidance regarding eating and drinking, including the balance of cost and benefit, the burden, to allow for breaking of dietary laws [22,24,25].

Food and drink are nourishment and hydration but also they are highly symbolic. This symbolism is often forgotten in the clinical realm, but it is equally important to the individual and their religious beliefs. The Scottish government guidance on *Food in Hospitals* comprehensively addresses extensive factors, including special populations who often have poorer access to health care. There is a clear separation of: “*‘Special diets’ refer to those meeting cultural or religious needs, while ‘personal diets’ are those meeting personal preferences.*” [26] (p. 112). This separation we take as an acknowledgement that religious tenets addressing food are more than a *nice-to-have* preference. The document details the dietary rules of five cultural groups and nine religions, including issues around when fasting is important—an often-forgotten aspect of nutritional information see Figure 1 [26].

Thus, to support the person with EDS difficulties we need to be aware of broad religious preferences and our specific patient’s interpretation of their faith. Starting from the patient/family perspective is an efficient way to pinpoint what matters and what does not. Negotiated information sharing, as with all areas of care, requires professionals to help the person work out the questions they need to ask. General knowledge of what might be important in terms of religion and dietary issues is a useful start. Rituals, sacraments, and special meals are often more related to context than food content. For example, in Roman Catholicism, the *host* is the wafer that has become the body of Christ in the sacrament of Holy Communion:

“*Context allows us to distinguish the host from ordinary food before we even see the actual host we are about to swallow. Were a sanctified host inserted into a layer of lasagna [sic] in a restaurant at lunch hour, we would not perceive it as an essentially different presence. Our intellects do not necessarily perceive something about the host itself that allows us to recognize it as Eucharist and not ordinary food. We get this because of the context of the rite.*” [22] (pp. 261–262).

Addressing issues related to religion and EDS do not necessarily require major system change. For example, with some cultures, the temperature of food or drink is important and so offering drinks of the appropriate temperature, e.g., hot tea instead of cold water at meals, may make a huge difference to the comfort of a patient [27]. Certain foods might be offered at meals, such as yoghurt, which relates to the sacredness of the cow in Hindu belief. Observing a moment of reflection before eating, such as saying grace, or thanking the food, is a principle found in many religions. Where and how food is prepared is highly significant for some religious communities and may require formal certification. For example, very observant Jews will require “Glatt” kosher food [23], which is prepared under the supervision of kashrut certification agencies (e.g., London Beth Din [28]). Unless this certification is clear (hechsher certified), the patient and/or family may not eat or drink. This knowledge may impact engagement with EDS activities. These issues are already being addressed in the world of pharmaceuticals, where the same dietary principles apply [29].

As clinical professionals, we may be the first people that patients or families discuss EDS difficulties with, and the potential impact on their religious experiences. If people have not faced EDS problems before, they may never have considered the nuances such as not necessarily having to be in a particular building, or have a specific food, in order to experience meaningful religious events. Close relations with the chaplaincy service and local religious leaders are essential to support this component of care.

## 5. The Burden of Rules and the Shadow Side of Misinterpretation

The “need to feed” is a universal driver which is brought acutely into focus when EDS becomes difficult/impossible for an individual. The communal nature of EDS impacts all stakeholders and decision-makers. Considering the individual nature of the religious experience, it can be challenging to know who to listen to in terms of weighting decisions: the family, the religious leader, or overarching religious tenets, which may be interpreted differently at a local level. Particularly when the patient does not have decisional capacity, we may feel the need to consider who we are treating, and be alert for possible coercion [30].

Variation in the interpretation of religious laws and guidance by religious leaders may have real influence over decisions made by patients, family, and staff [25,31]. Understanding this tension needs careful consideration and awareness. This is particularly important if patients are not able to make decisions for themselves and have not previously expressed preferred choices. Preferences might include personal religious devoutness/adherence in relation to EDS and other health-related decisions. The level of religious adherence to dietary requirement may differ between the patient, family, religious leader, and staff. Balancing the needs of all stakeholders in this dynamic is challenging. The locus of ultimate decision-making may not be with the healthcare team or even the patient, depending on cultural perspective [32]. This may be either positive and supporting patient-centered care or it could sit uncomfortably with some of the stakeholders. Good mediation is skilled and essential work for the multidisciplinary team. Having current evidence-based information related to religious consensus may be a useful output. Involving chaplaincy services and the religious establishments across the UK will add value to local discussions [19].

Even when resources are available, people may not want to access them for fear of being seen as failing in family duties [5] or spiritual challenge [6]. The EDS impairments faced by a person can also have profound effects on family members and their religious experience [33]. Socioeconomic status and access to health care is also affected by cultural grouping, but this is subtle and complex rather than a cause-and-effect issue [4].

## 6. Discussion

There are many facets to a person’s identity, and religion is one of those. Increasing realization of the link between health care and religion calls for those concerned with the religious/faith/spiritual component to be recognized as central to the team and professionally accountable [4,19,20,34]. These important reports and guidelines sadly lack reference to the impact that EDS difficulties may have on the person’s experience of their religion and vice versa. We need to share learning about EDS and associated difficulties with religious groups. Equally, we need to learn from them about how best to support the spiritual aspect of care in health and illness. To optimize the needs of the person at the center of care, we need to ensure that the multidisciplinary team includes the patient and family, and appropriate religious expertise which may come from a range of informed sources. A multi-pronged approach also needs action at the level of service delivery: engagement between health teams and religious experts would provide for guidelines, policies, and changes to food services. Increasing the overarching awareness and support for EDS issues will impact the experience for all patients.

Future research and opportunities for continuing professional development should address how EDS issues affect the religious experiences of people. Thinking about major religious festivals in the world, e.g., Eid, Diwali, Easter, Passover, Ugadi, and so forth, all involve food and drink in some form or absence. How might a person’s religious identity be affected by the limitations of tube feeding? Our health and care systems need a better understanding of the drivers behind an individual’s religious perspective and how to manage their EDS difficulties. This would reduce the likelihood of heart-wrenching discussions regarding spiritual wellbeing occurring in times of crisis.

## 7. Conclusions

Eating, drinking, and swallowing are essential for human existence, but are so much more than mere fuel consumption. Given the significant impact of religious belief on people’s experiences and health outcomes, we need to address this early in the patient’s journey. Appreciating the significance of the interplay of EDS and religious belief will help to promote optimal engagement with health care resources. If we raise the topic explicitly then patients, families, and professionals will know which questions to ask. Early identification of potential EDS issues and clear discussions might avoid heartache for all stakeholders involved. Health care services are starting to address this, and to develop resources we should engage with our patients and religious experts. This is very much a mutual sharing of wisdom which will lead to efficient service delivery and compassionate care.

## Figures and Tables

**Figure 1 geriatrics-07-00041-f001:**
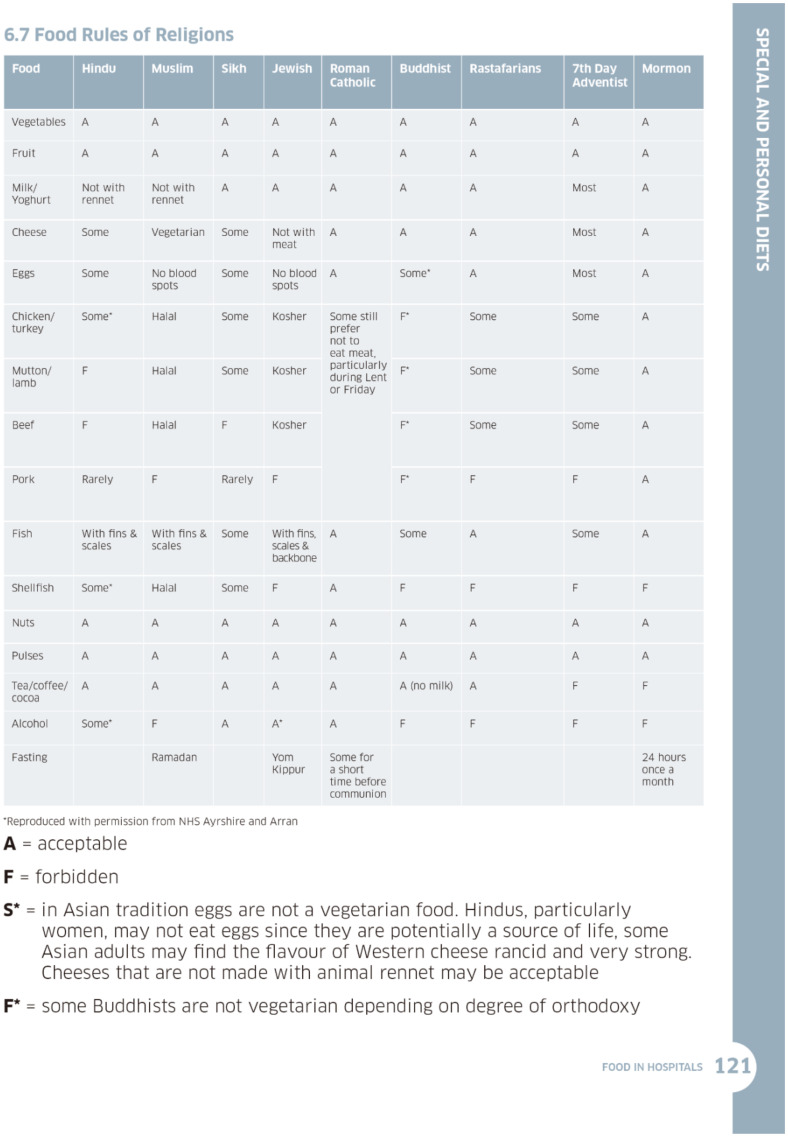
Food Rules of Religions (Food in Hospitals, Scottish Government (2016)).

## Data Availability

Not applicable.

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
