# Peer review of "Eating, Drinking, and Swallowing Difficulties: The Impacts on, and of, Religious Beliefs"

_geriatrics, 2022, doi:10.3390/geriatrics7020041_

Round 1
Reviewer 1 Report
Thank you for the opportunity to read this very relevant piece. I think it is of interest to readers and has merit in this journal.
Specific points:
My first thought on reading about 'chaplaincy services' is that they are Christian, it might be useful to use a more obviously multi-faith term, or to say that chaplaincy services are typically now multi-faith/ multi-philosophy.
p2. line 44. Are you suggesting what we eat/ don't eat is always a matter of choice in a philosophical way? - I'm thinking about people with food allergies or metabolic issues or Ehlers Danlos for example.
p2. Line 54 - There is a typo -"a practical guided".
line 158 - I'm pleased to see here reference to spirituality as it covers a larger understanding of faith that is not captured by religion, or doesn't come under a named religion. It also aligns with the fourth dimension of health from WHO. I think earlier reference to spirituality would help the reader understand the topic and make clearer whether you are talking about spiritual health or not.
line 193 - typo "staring"
top of page 5 - I think this is supposed to be a list? it doesn't read right.
line 200 - I had to re-read this several times as I was not familiar with the term 'host'. I was thinking of the person who hosts a dinner party for example - a good example of misunderstanding! Perhaps a quick definition of 'host' would be helpful as a precursor.
line 207 - grammar here needs fixing.
line 218 - typo "principle"
line 221 - who are the 'we' being referred to here? This sentence seems to combine two important factors that are separate - one being who the patient/ families talk to, the other being that families can negotiate how their religious preferences/ needs can be addressed to their satisfaction.
line 292 - typo "religions"
Discussion para 1 - I think there is a multi-pronged approach suggested. At the level of service delivery, engagement between health teams and religious experts would provide for guidelines, policies, changes to food services and more overarching awareness and support that can impact the experience for all patients. At the level of the patient, including a religious guide/ adviser/ expert would be useful for specific individuals and their family. This para seems to suggest a religious expert needs to be involved with every patient - or maybe your suggestion of 'religious expertise' could be described a little more.
I like your use of the words 'heart wrenching' and 'heartache'. They are appropriately emotive but maybe need to be linked back to matters of spiritual wellbeing for those who are more 'medical' minded.
Author Response
Please see the attachment box, thank you.

Reviewer 2 Report
The paper is a short communication comprising a position paper on bioethics and patients’ care / health systems' current practices regarding eating, drinking, and swallowing difficulties in relation to religious beliefs. This is a matter rarely examined and very relevant in the scope of holistic patient approach and commitment to inclusivity at health care.
Some points that the authors should address
Title “Eating, drinking, and swallowing difficulties: the impact on and of religious beliefs” rephrase please
Introduction
Page 2 line 54 “Religion or belief: a practical guided for the NHS”correct to “Religion or belief: a practical guide for the NHS”
Page 5 line 195“Negotiated information sharing as with all areas of care, helping the person work out the questions they need to ask” rephrase please
Author Response

(The authors gave the same response as above.)
